# Journey to facility birth in Zanzibar: a questionnaire-based cohort study of patients' perspectives on preparedness, access and quality of care

Tanneke Herklots ,[1] Lara D'haene,[2] Khairat Said Mbarouk,[3] Mubina Rajhy,[3] Simone Couperus,[4] Tarek Meguid,[5] Arie Franx,[1] Maria P H Koster,[1] Benoit Jacod[6]

► Additional material is published online only. To view please visit the journal online (http://dx.doi.org/10.1136/bmjopen-2020-040381).

[1]Obstetrics & Gynaecology, Erasmus MC, Rotterdam, The Netherlands
[2]Emergency Medicine, Hospital Doctor Jose Molina Orosa, Arrecife, Lanzarote, Spain
[3]Obstetrics & Gynaecology, Mnazi Mmoja Referral Hospital, Zanzibar, Tanzania
[4]Obstetrics & Gynaecology, University of Utrecht Faculty of Medicine, Utrecht, The Netherlands
[5]Village Health Works, Kigutu, Burundi
[6]Obstetrics & Gynaecology, Onze Lieve Vrouwe Gasthuis, Amsterdam, The Netherlands

**Correspondence to**
Dr Tanneke Herklots;
t.herklots@erasmusmc.nl

## ABSTRACT

**Introduction** Tackling substandard maternity care in health facilities requires engaging women's perspectives in strategies to improve outcomes. This study aims to provide insights in the perspectives of women with severe maternal morbidity on preparedness, access and quality of care in Zanzibar's referral hospital.

**Methods** In a prospective cohort from April 2017 to December 2018, we performed semistructured interviews with women who experienced maternal near-miss complications and matched controls. These focused on sociodemographic and obstetric characteristics, perceived accessibility to and quality of facility care with 15 domains, scored on a one-to-five scale. Participants' comments and answers to open questions were employed to illustrate quantitative outcomes. Zanzibar's Medical Research and Ethics Committee approved the study (ZAMREC/0002/JUN/17).

**Results** We included 174 cases and 151 controls. Compared with controls, patients with a near-miss had less formal education (p=0.049), perceived their wealth as poor (p=0.002) and had a stillbirth more often (p<0.001). Many experienced a delay in deciding to seek care. More than controls, near-miss patients experienced barriers in reaching care (p=0.049), often of financial nature (13.8% vs 4.0%). Quality of care was perceived as high, with means above 3 out of 5, in 14 out of 15 domains. One-fifth had an overall suboptimal experience, mostly regarding informed choice and supplies availability. Additional comments were expressed by a minority of participants.

**Conclusion** Most patients promptly sought, accessed and received maternity care in Zanzibar's referral hospital. A minority experienced barriers, mostly financial, in reaching care and more so among patients with near-miss complications. Quality of facility care was generally highly rated. However, some reported insightful critical perceptions. This study highlights the impact of sociodemographic differences on health, the value of involving patients in decisions regarding maternity care and the need to ensure availability of medical supplies, all which will contribute to improved maternal well-being.

### Strengths and limitations of this study

► The study involved a wide range and large number of maternal patients in Zanzibar's main hospital.
► The given comments reached outcome saturation, for which we trust them to experienced reality, thus being valid starting points for healthcare improvement to provide to policy-makers.
► This study displays how facility maternity care is perceived by the patients themselves, and does so in an extensive, profound and both quantitative and qualitative manner.
► Because we could not include women who had not reached the hospital, nor patients who were not in possession of a mobile phone, the study had a selection bias potentially at the cost of those suffering more from poverty.
► There might have been a certain level of information bias because the (majority of) interviews was conducted in the hospital, potentially discouraging people to criticise the facility they were at.

and mortality. Efforts have gone into ensuring availability of essential interventions, promoting facility deliveries and skilled birth attendance.[1] Unfortunately, despite a widespread increase in facility deliveries, the improvement in pregnancy outcomes has been slower than expected. This might be because women and their families are not prepared in time to seek facility care or face limitations in accessing facility care, but there seems to be a causal role for the low quality of care provided in many facilities.[2] Low quality of care negatively affects the willingness to seek facility care, either in a future pregnancy or of other pregnant women in the community. Poor quality of care is not limited to but includes maltreatment during childbirth, which has been found to be common in many low-income settings.[3–5] Tackling substandard maternity care requires incorporating the

## INTRODUCTION

Low-income countries carry a disproportionately large burden of maternal morbidity

perspectives of women in designing strategies to improve maternal and child outcomes.

A specific group of women within the healthcare system are those who experience life-threatening complications during pregnancy and childbirth, so-called maternal near-miss (MNM) complications, but survive. Maternal deaths are an often-used proxy for assessing the quality of care but in the group of all childbearing women, represent an absolute small number. Women who had an MNM (from now on referred to as MNM) represent a larger proportion, while still being a group small enough to accommodate in-depth assessments of their multifacete experiences in the healthcare system.

In Zanzibar, Tanzania, around half of the deliveries take place at home and half in a health facility.[6] Of the latter, around 30% take place in the region's main public referral health facility: Mnazi Mmoja Hospital (MMH). The maternal mortality rate at MMH is around 400 per 100 000 livebirths.[7] Multiple quality of care assessments have been performed at Tanzanian healthcare facilities guided by the WHO Standard for Care.[8] In 2012, a large unmet need for basic and comprehensive emergency obstetric and neonatal care was found throughout Zanzibar, including shortages of staff and supplies, and an insufficient geographical access.[9] In 2016, an assessment in Zanzibar showed that five main governmental hospitals, including MMH, were overall not conducive for high-quality maternal and child healthcare provision (Meguid T, Mrisho Haji F, Herklots T, Rajhy M, Kassim NI. Conduciveness for High Quality of Maternal and Child Health Care in Hospitals in Zanzibar. 2016). In 2017, an assessment of 37 maternity units in Zanzibar showed how more than half of those had challenges in providing hand and surface hygiene.[10]

This objective of this study is to explore the experiences of women with and without near-miss complications regarding their preparedness for, access to and quality of facility care in Zanzibar's main hospital. Understanding maternity care with its qualities and flaws from the patients' perspectives can inform local and regional maternal health policies and research agendas.

## METHODS

From April 2017 to December 2018, we conducted a prospective cohort study at the department of Obstetrics and Gynaecology of MMH. We performed semistructured interviews with women who experienced an MNM and control participants. MNM were identified during the department's twice daily meetings by a locally adapted and validated version of the WHO near-miss approach (for criteria see online supplemental table 1).[7] Controls, matched on a 1:1 ratio, did not have a severely complicated pregnancy and had a similar date of admission (up to 3 days before or after the MNM's admission date), similar mode of delivery (vaginal, by caesarean section or instrumentally assisted) and similar gestational age (first

or second trimester or third trimester preterm or term) as MNM.[11]

In addition to providing informed consent, inclusion criteria were age of 18 years or above, no severe prediagnosed psychiatric disorders, residence on Unguja (Zanzibar's main island on which MMH is located) and to be reachable by mobile phone. Informed consent of all participants was obtained in writing or verbally, in case of illiteracy. An interview was performed before discharge or, in case of logistic restrictions, within 1 week of discharge at the participant's home. The interviews were performed by a researcher and a mediator/translator, both not involved in the clinical care of the woman. The interview comprised the following: sociodemographic characteristics (age, self-reported level of formal education, type of occupation, marital status, perceived wealth), obstetric characteristics (parity—with a grand multiparity defined as a parity above 4, singleton or multiple pregnancy, gestational age, planned location for delivery and pregnancy outcome being abortion, live birth or stillbirth), utilisation and opinion of antenatal care services (not included in this publication), perception on the accessibility of facility care and perception on the quality of facility care (see online supplemental table 2, for the interview outline). We decided to combine quantitative and qualitative outcomes. Employing quantitative measurements of experiences around child birth and satisfaction with healthcare are in line with recent efforts to design validated standard outcome measurements sets, including patient-reported outcomes, such as the Pregnancy and Childbirth set of the International Consortium for Health Outcomes Measurement.[12] A quantitative approach allows for easier comparison between categories, across settings and over time. While discussing those experiences on accessibility to and quality of care, the interviewer requested interviewees' comments in addition to the quantitative scores, which yielded a wealth of quotes.

SPSS Statistics (V.25) and OpenEpi (V.3.01) were used for statistical analysis. Numeric outcomes were categorised similarly as in previous studies in this setting: age younger than 20 years, 20–35 years and older than 35 years; and parity zero, one to four or higher than four.[7 13] Baseline characteristics were compared using Pearson's $\chi^2$ tests and, in case of small sample sizes, Fisher's exact test. Regarding quality of care, the interview outcomes were on a 5-point scale, from which means per question were calculated and compared between MNM and controls using Mann-Whitney U tests. Suboptimal care was defined as a score of 3 or lower. Proportions of participants grading the assessed element as suboptimal were reflected in percentages. ORs with 95% CIs were calculated to compare between MNM and controls, to assess if having had a pregnancy with a life-threatening complication affects quality of care perception. We observed a discrepancy between the fact that the majority of patients perceived quality of care as good or very good and the fact that give comments most often had a negative connotation. To investigate this discrepancy further, we

compared the frequency of negative comments in participants having scored as positive or as suboptimal on the domain to which the comments pertain. We deemed the number of participants giving comments to be sufficiently high to perform this analysis on four domains: information, informed choice, respect and privacy.

A p<0.05 was considered statistically significant. Quotations taken from participants' comments and answers to open questions were employed to illustrate the quantitative outcomes.

### Patient and public involvement

The lack of patient's perspectives in maternal health research in our setting motivated our research questions and methodology. Patients and public were not directly involved in the design of the study, nor in the plans for dissemination. The flexible construct of the interviews did allow the patient to decide on which topics she wanted to elaborate. Dissemination of the study outcomes will, next to publication in an open access scientific journal, take place in the study setting, to the index hospital and to Zanzibar's Ministry of Health. We hope this study can inspire the local research agenda, motivating patient involvement.

### RESULTS

During the study period, 269 patients were identified as MNM, of whom 174 (64.7%) were interviewed. Reasons for no study participation were patients' personal reasons, meeting one of the exclusion criteria or an unexpected early discharge home. A total of 151 control participants were interviewed.

### Participants' characteristics

Baseline and outcome characteristics are displayed in table 1. MNM and controls were comparable at baseline regarding age, occupation and marital status. MNM were significantly more likely to have completed primary education only (29.9% vs 19.2%, p=0.05), more often perceived their own wealth as very poor or poor (19.0% vs 7.3%, p=0.002), were less often nulliparous (32.2% vs 50.3%, p=0.002), and more often grand multiparous (13.9% vs 6.0%, p=0.002). Of the MNM pregnancies, 42.9% ended in a stillbirth compared with 5.4% of control pregnancies (p<0.001). Nearly two-thirds of all participants had initially planned to deliver at MMH, controls significantly more so than MNM (total 66.5%; 58.6% MNM vs 75.5% controls, p=0.004).

### Access to care

Motivations for choosing a delivery location did not differ between MNM and controls and were most often because of previous experience and expectations (36.6%) or medical indication (28.0%).

> I didn't decide yet in which hospital but I don't want to deliver at home because you can have a problem and not be able to solve it—MNM participant, 25 years old, first pregnancy

The majority of MNM (59.2%) initiated their journey to MMH because they recognised that their condition was deteriorating, while less than a quarter (23.0%) of them said they were unaware.

> I recognized it was not good. I thought I needed to have an operation because that is what happened to my sister-in-law.—control participant, 36 years old, fourth pregnancy

Table 2 displays the various elements on access to care investigated. Over half of the MNM (55.6%) and the controls (66.1%) said to have taken more than 2 hours before they decided to seek medical care after starting to feel sick or labour started (p=0.18). After that, it took 36.2% of MNM and 29.1% of controls longer than 30 min to reach a health facility (p=0.10). Once arrived, the vast majority was attended to promptly, but some had waited longer than 3 hours, and MNM significantly more often than controls (9.3% vs 5.4%, p=0.04).

Close to one-fifth of all interviewees (18.6%) experienced the access to care as suboptimal, with a significant higher number of MNM than controls reporting to have encountered problems of financial, logistical or referral-related nature (22.4% MNM vs 13.9% controls, p=0.049). Many interviewees shared that they had struggled with finding and paying petrol, a borrowed car or public transportation.

> I had to wait and find a bus for six hours. There should be transport available—MNM participant, 19 years old, second pregnancy

> I came by ambulance from [my area] and paid 10,000TSH (equivalent to 5 USD, authors). I find it not fair of them that I had to pay. What if I had no money? —MNM participant (area of residence to MMH around a 20-minutes-drive, 8–9 km), 40 years old, sixth pregnancy

> Because my husband was not able to buy petrol for our car, I took a dala dala (public mini bus, authors) for three hours from [my area] to MMH.—MNM participant (area of residence to MMH around one and a half hour-drive by car, 40km), 38 years old, eight pregnancy

### Quality of care

Table 3 displays the means of the outcomes per interview question reflecting quality of care. Most of these given scores are high, that is, above four. Lowest scores regard having an informed choice (total group 2.67, MNM 2.49 and controls 2.86) and being informed (total group 3.87, MNM 3.61 and controls 4.17), including being informed about the death of their baby (total group 3.90, MNM 3.89 and controls 4.00). Mean differences between MNM and controls were statistically significant for some topics, such as having felt informed and perception of motivation

**Table 1** Baseline and outcome characteristics

| | Total N (%) | MNM N (%) | Controls N (%) | P value |
|---|---|---|---|---|
| Total | 325 (100) | 174 (53.5) | 151 (46.5) | |
| Age | | | | 0.25 |
| Data available* | 324 (99.7) | 174 (100) | 150 (99.3) | |
| <20 years | 16 (4.9) | 11 (6.3) | 5 (3.3) | |
| 20–35 years | 267 (82.4) | 138 (79.3) | 129 (86.0) | |
| >35 years | 41 (12.7) | 25 (14.4) | 16 (10.7) | |
| Parity | | | | 0.002 |
| Data available* | 324 (99.7) | 173 (99.4) | 151 (100) | |
| 0 (nulliparous) | 132 (40.7) | 56 (32.4) | 76 (50.3) | |
| 1–4 (multiparous) | 159 (49.1) | 93 (53.8) | 66 (43.7) | |
| 5 or more (grand multiparous) | 33 (10.2) | 24 (13.9) | 9 (6.0) | |
| Formal education level | | | | 0.049 |
| No formal education | 24 (7.4) | 15 (8.6) | 9 (6.0) | |
| Primary | 81 (24.0) | 52 (29.9) | 29 (19.2) | |
| Secondary | 194 (59.7) | 97 (55.7) | 97 (64.2) | |
| Higher | 26 (8.0) | 10 (5.7) | 16 (10.6) | |
| Occupation | | | | 0.80 |
| Paid work or self-employed | 148 (45.5) | 76 (43.7) | 72 (47.7) | |
| Housework | 163 (50.2) | 91 (52.3) | 72 (47.7) | |
| Non-paid work or student | 9 (2.8) | 5 (2.9) | 4 (2.6) | |
| Unemployed | 5 (1.5) | 2 (1.1) | 3 (2.0) | |
| Marital status | | | | 0.21 |
| Married | 299 (92.0) | 157 (90.2) | 142 (94.0) | |
| Unmarried | 26 (8.0) | 17 (9.8) | 9 (6.0) | |
| Perceived wealth | | | | 0.002 |
| Very poor or poor | 44 (13.5) | 33 (19.0) | 11 (7.3) | |
| Average | 280 (86.2) | 140 (80.5) | 140 (92.7) | |
| Rich or very rich | 1 (0.3) | 1 (0.6) | 0 (0.0) | |
| Singleton/multiple pregnancy† | | | | 0.03 |
| Data available | 261 (80.3) | 144 (82.8) | 117 (77.5) | |
| Singleton | 242 (92.7) | 129 (89.6) | 113 (96.6) | |
| Multiple | 19 (7.3) | 15 (10.4) | 4 (3.4) | |
| Planned location for delivery | | | | 0.004 |
| MMH | 216 (66.5) | 102 (58.6) | 114 (75.5) | |
| Other hospital | 97 (29.8) | 63 (36.2) | 34 (22.5) | |
| Health centre | 8 (2.5) | 6 (3.4) | 2 (1.3) | |
| Home | 4 (1.2) | 3 (1.7) | 1 (0.7) | |
| Pregnancy outcome | | | | <0.001 |
| Data available* | 319 (98.2) | 170 (97.7) | 149 (98.7) | |
| Ending before 20 weeks | 43 (13.5) | 25 (14.7) | 18 (12.1) | |
| Live birth | 191 (59.9) | 69 (40.6) | 122 (81.9) | |
| Stillbirth | 81 (25.4) | 73 (42.9) | 8 (5.4) | |
| Still pregnant at discharge | 4 (1.3) | 3 (1.8) | 1 (0.7) | |

*In case of missing data, the table shows the available data per category in the first row; the consequent proportions are calculated with the available data as denominator.
†This excludes participants who were admitted with a gestational age below 20 weeks, including pregnancies which ended in abortion and ectopic pregnancies.
MMH, Mnazi Mmoja Hospital; MNM, maternal near-miss.

**Table 2** Experience of accessibility of care

| | Total N (%) | MNM N (%) | Controls N (%) | P value* |
|---|---|---|---|---|
| Total | 325 (100.0) | 174 (53.5) | 151 (46.5) | |
| Sought care × time after complaints started (delay 1) | | | | 0.18 |
| Data available† | 284 (87.4) | 151 (86.8) | 133 (88.1) | |
| Immediately | 60 (21.1) | 40 (26.5) | 20 (15.0) | |
| <2 hours | 35 (12.3) | 19 (12.6) | 16 (12.0) | |
| 2–8 hours | 96 (33.8) | 45 (29.8) | 51 (38.3) | |
| >8 hours | 76 (26.8) | 39 (25.8) | 37 (27.8) | |
| Not applicable | 17 (6.0) | 8 (5.3) | 9 (6.8) | |
| Reached care × time after start of journey *(delay 2)* | | | | 0.10 |
| Data available† | 261 (80.3) | 129 (74.1) | 132 (87.4) | |
| <15 min | 70 (25.7) | 36 (26.1) | 34 (25.4) | |
| 15–30 min | 102 (37.5) | 43 (31.2) | 59 (44.0) | |
| 30–60 min | 49 (18.0) | 24 (17.4) | 25 (18.7) | |
| >60 min | 40 (14.7) | 26 (18.8) | 14 (10.4) | |
| Received care × time after arrival in health facility (delay 3) | | | | **0.04** |
| Data available† | 304 | 158 | 146 | |
| Immediately | 251 (78.7) | 122 (71.3) | 129 (87.2) | |
| <3 hours | 29 (9.1) | 20 (11.7) | 9 (6.1) | |
| >3 hours | 24 (7.5) | 16 (9.3) | 8 (5.4) | |
| Experienced access to care | | | | 0.48‡ |
| Data available† | 322 | 171 | 151 | |
| Very poor | 2 (0.6) | 2 (1.2) | – | |
| Poor | 10 (3.1) | 5 (2.9) | 5 (3.3) | |
| Not poor, not good | 48 (14.9) | 28 (16.4) | 20 (13.2) | |
| Good | 59 (18.3) | 35 (20.5) | 24 (15.9) | |
| Very good | 203 (63.0) | 101 (59.1) | 102 (67.5) | |
| Encountered problems | | | | **0.049§** |
| None | 265 (81.5) | 135 (77.6) | 130 (86.1) | |
| Any | 60 (18.5) | 39 (22.4) | 21 (13.9) | |
| Financial | 30 (9.2) | 24 (13.8) | 6 (4.0) | |
| Referral from one health facility to another | 10 (3.1) | 6 (3.4) | 4 (2.6) | |
| Logistics | 20 (6.2) | 9 (5.2) | 11 (7.3) | |

*P values are calculated with $\chi^2$ tests excluded the row of 'data available'.
†The proportion of data available is calculated, the proportion of the following subcategories are calculated with the available data as the denominator.
‡This p value has been calculated with combining the subcategories 'very poor' and 'poor'.
§This p value has been calculated by comparing the categories 'none' and 'any'.
MNM, maternal near-miss.

of staff, but reflect small differences. One participant summarised her experience as follows:

I am very satisfied because I am alive—MNM participant, 28 years old, first pregnancy

As displayed in table 4, a total of 70 interviewees (21.5%), comprising 43 MNM (24.7%) and 27 controls (17.9%), perceived the overall experience of the care during their admission as suboptimal. Most participants

rated informed choice poorest, being scored as suboptimal by 57.5% of all participants. This is followed by sufficiency of supplies (28.3%), being informed in general (28.3%), and being informed in case of death of their baby (25.9%).

If I had money I would go to a private hospital. The first doctor after the operation didn't tell me my baby died. … I felt ignored. …They didn't tell me anything.

**Table 3** Means on a 5-point scale (1=very poor, 2=poor, 3=not poor, not good, 4=good, 5=very good) and mean differences

| | Total Mean (SD) | MNM Mean (SD) | Controls Mean (SD) | Mean difference (95% CI) |
|---|---|---|---|---|
| Did you feel treated with respect? | 4.54 (0.80) | 4.50 (0.85) | 4.58 (0.74) | −0.07 (−0.25 to 0.10) |
| Did you feel informed about treatments/complications? | 3.87 (1.54) | 3.61 (1.62) | 4.17 (1.40) | −0.55 (−0.89 to 0.22) |
| Did you feel you had an informed choice in the services you received? | 2.67 (1.80) | 2.49 (1.76) | 2.86 (1.84) | −0.37 (−0.76 to 0.03) |
| Did you feel you had privacy? | 4.47 (0.96) | 4.36 (1.12) | 4.59 (0.72) | −0.23 (−0.44 to 0.02) |
| Were you able to receive emotional support from relatives/friends? | 4.47 (0.97) | 4.51 (0.91) | 4.42 (1.03) | 0.09 (−0.12 to 0.30) |
| Did you feel you were provided with emotional support by the staff? | 4.27 (1.18) | 4.23 (1.20) | 4.32 (1.16) | −0.09 (−0.35 to 0.17) |
| Did you feel that the staff was motivated and available? | 4.41 (0.88) | 4.32 (0.95) | 4.52 (0.78) | −0.20 (−0.40 to 0.01) |
| Did you feel safe in the hospital? | 4.65 (0.67) | 4.63 (0.69) | 4.66 (0.64) | −0.03 (−0.18 to 0.11) |
| Did you feel the health workers took the time for you? | 4.51 (0.83) | 4.47 (0.87) | 4.56 (0.78) | −0.08 (−0.26 to 0.10) |
| Did you experience financial barriers to getting the right treatment? | 4.59 (0.88) | 4.47 (1.00) | 4.74 (0.70) | −0.27 (−0.46 to 0.08) |
| Did you think the hospital had sufficient supplies to care for you? (Sanitation, medicine, equipment) | 4.01 (0.99) | 3.91 (0.97) | 4.10 (1.02) | −0.17 (−0.39 to 0.05) |
| How do you feel about the way your baby was treated? | 4.67 (0.77) | 4.64 (0.87) | 4.69 (0.70) | −0.05 (−0.28 to 0.18) |
| How do you feel about the overall experience of your admission? | 4.33 (0.90) | 4.24 (0.93) | 4.42 (0.86) | −0.19 (−0.38 to 0.01) |
| In case baby died: how do you feel about the way you were informed on the death of your baby? | 3.90 (1.54) | 3.89 (1.56) | 4.00 (1.48) | −0.11 (−1.08 to 0.85) |
| In case baby died: how do you feel about the way you, your baby and the baby's father were treated after your baby's death? | 4.16 (1.39) | 4.31 (1.26) | 3.27 (1.85) | 1.04 (0.16 to 1.91) |
| Would you recommend your sister/friend to deliver in the same place? | **Total N (%)** | **MNM N (%)** | **Controls N (%)** | |
| Yes | 309 (95.1) | 164 (94.3) | 145 (96.0) | 0.85 (0.57 to 1.26) |
| No | 16 (4.9) | 10 (5.7) | 6 (4.0) | |

MNM, maternal near-miss.

I am heartbroken here.—MNM participant, 31 years old, second pregnancy

The odds of experiencing care as suboptimal were significantly higher for MNM concerning financial barriers in receiving high quality care (OR 2.76, 95% CI 1.34 to 5.70), privacy (OR 2.32, 95% CI 1.14 to 4.72), information provision (OR 2.09, 95% CI 1.26 to 3.46) and availability and motivation of staff (OR 1.94, 95% CI 1.07 to 3.54).

They didn't tell me anything. I had no choice. I didn't feel extremely safe because some patients died. There is not enough medicine in the hospital and some check-ups needed to be done outside of the hospital, this costs a lot of money. – control participant, 23 years old, second pregnancy

I don't know how my baby was treated. I haven't seen my baby yet at all. I don't know why, they told me I can't see him yet. I want to see him. I believe I will see him.—MNM participant, 29 years old, third pregnancy

Figure 1 displays additional analysis on four domains, showing the proportion of participants who commented on the matter, among those who gave a high score and a suboptimal score, respectively. However, less than 10% of the participants provided any comments, except on informed choice where almost 20% of them did. Participants scoring the quality as good or very good were just as likely as participants scoring quality as suboptimal to give negative comments when asked to answer freely.

Figure 1 good and suboptimal scores on domains of information provision, having an informed choice, having privacy and feeling to have been treated with respect, and the proportions of participants in both groups who commented on the index domains.

The following quotes further exemplify how participants who gave high quantitative scores did have critical comments on those topics:

**Table 4** Suboptimal scores on quality of care reflecting absolute numbers and the proportion of the group of participants who answered with 1, 2 or 3

| | Total, N (%) | MNM, N (%) | Control, N (%) | OR (95% CI) |
|---|---|---|---|---|
| Total | 325 (100.0) | 174 (53.5) | 151 (46.5) | – |
| Overall experience | 70 (21.5) | 43 (24.7) | 27 (17.9) | 1.51 (0.88 to 2.59) |
| Informed choice | 187 (57.5) | 108 (62.1) | 79 (52.3) | 1.49 (0.96 to 2.32) |
| Sufficient supplies | 104 (32.0) | 61 (35.1) | 43 (28.5) | 1.36 (0.85 to 2.17) |
| Information | 92 (28.3) | 61 (35.1) | 31 (20.5) | *2.09 (1.26 to 3.46)* |
| Emtional support staff | 66 (20.3) | 37 (21.3) | 29 (19.2) | 1.14 (0.66 to 1.96) |
| Availability and motivation of staff | 57 (17.5) | 38 (21.8) | 19 (12.6) | *1.94 (1.07 to 3.54)* |
| Time for patient | 43 (13.2) | 22 (12.6) | 21 (13.9) | 0.90 (0.47 to 1.70) |
| Financial barriers | 42 (12.9) | 31 (17.8) | 11 (7.3) | *2.76 (1.34 to 5.70)* |
| Privacy | 41 (12.6) | 29 (16.7) | 12 (7.9) | *2.32 (1.14 to 4.72)* |
| Emotional support family | 39 (12.0) | 20 (11.5) | 19 (12.6) | 0.90 (0.46 to 1.76) |
| Respect | 36 (11.1) | 22 (12.6) | 14 (9.3) | 1.42 (0.70 to 2.88) |
| Feeling of safety | 18 (5.5) | 10 (5.7) | 8 (5.3) | 1.09 (0.42 to 2.84) |
| Treatment of the baby | 10 (3.1) | 4 (5.8) | 6 (4.9) | 0.57 (0.16 to 2.05) |
| | | **Total n=73** | **Total n=12** | |
| In case baby died: information | 22 (6.7) | 20 (27.4) | 2 (16.7) | *9.68 (2.22 to 42.11)* |
| In case baby died: supportive treatment | 13 (4.0) | 9 (12.3) | 4 (33.3) | 2.01 (0.60 to 6.65) |

OR calculated for MNM versus controls.
MNM, maternal near-miss.

Someone said to me: stay where you are or I will beat you, while I was in labour. When I see that doctor now, I feel very uncomfortable.—MNM participant, 30 years old, second pregnancy >scored a 4 on 'respect'

No privacy because even delivery was open and everyone could see—MNM participant, 31 years old, seventh pregnancy >scored a 5 on 'privacy'

Staff should change their attitude. There should be better rules/guidelines about how to treat patients. They should treat patients with more respect and listen to us. They should be more focused.—MNM

participant, 24 years old, second pregnancy >scored a 4 on 'respect'

## DISCUSSION
### Main findings
On their journey to facility care, women who suffered from life-threatening complications and those who did not, equally went through extensive processes preceding the decision to go to the healthcare facility. Around one-fifth of them, MNM significantly more often so, subsequently experienced barriers, mostly due to financial constraints. Once in the hospital, care was given quickly. The quality of care reported through quantitative questionnaires was experienced as high. One-fifth of all participants, MNM significantly more often than controls, had an overall suboptimal experience with informed choice and availability of supplies scored lowest. Our expectation was that participants would more often comment negatively if they had given lower scores but this did not occur consistently. Generally, there was a positive quantitative assessment with critical comments expressed by a minority of the participants.

### Interpretation
In our study population, women who suffered from MNM reported to have experienced financial barriers more often than the controls did. Previous work found that financial costs were a major obstacle in seeking non-emergency obstetric care, but not so much in emergency

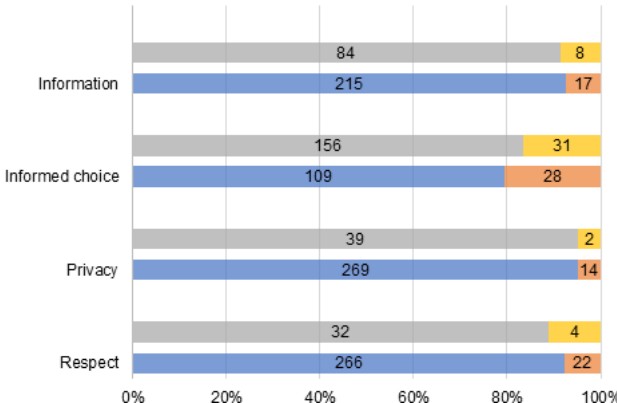

**Figure 1** Legend: top bar left side=suboptimal score with no comment; top bar right side=suboptimal score with Comment; bottom bar left side=good score with no Comment; bottom bar right side=good score with comment.

obstetric care.[14 15] We have insufficient insights into the course of disease to conclude whether these obstacles are causally related with outcomes. Though, they might reflect how a deprived socioeconomic situation creates a predisposition for poorer health outcomes. This is underscored by the fact that our study participants, who developed the near-miss complications, had completed lower levels of formal education and perceived their own wealth more often as poor. Additional research should be done to explore the nexus between economic factors and healthcare-seeking behaviour, like the studies from multiple countries in sub-Saharan Africa and from Bolivia which have displayed similar difficulties with poor information provision and consequent limitations in making informed choices as experienced by our study participants.[16–20]

Our study has shown that, severe obstetric complications do not seem to significantly impact patient's perception of care, either negatively or positively. Adverse outcomes will in most cases have led to intensification of clinical care, which could partly explain a positive experience from patients who had severe complications. On the other hand, possible negligence or poor communication and information provision have likely influenced perceptions negatively.

This study's methodology was mainly quantitative, but encouraged comments from the participants to enrich the numeric outcomes. Through calculations of averages we get a sense of experienced quality of care, but the valuable information for policy change really seems to come from the positive and negative extremes. The individual participants' comments reflected those extremes best, with their experiences of substandard care highlighting ample opportunities for improvement. This is most visible in their perception of quality of care which had a discrepancy between the quantitative and qualitative outcomes. We hypothesise that quantitative outcomes mainly reflect the norms in the healthcare system and broader society, with people adjusting their standards to what is currently available to them. Even so, qualitative data did shed a bright light on personal needs and preferences deviating from the current norm. The norm might be what it is, but that does not mean that the patient, who is depending on the healthcare system, finds the norm acceptable. For example, the vast majority scored 'privacy' as good or very good, while 1 in every 20 participants shared how they had to share beds, there was a lack of curtains and physical examinations were executed without shielding. The quotes reproduced in the article illustrate stories on poor treatment during hospitalisation that are similar to those of other women around the world,[14–18] whose expectations of smooth access, privacy, adequate provision of information, attitude and availability of supplies reflect essential elements of high-quality maternal care as standardised by WHO.[8] Dissatisfaction on not meeting these expectations might not be reflected directly given the propensity to normative answers in most, but can negatively impact future service utilisation and, consequently, maternal health outcomes.

Our study exemplifies the need for both quantitative and qualitative methods to get a genuine impression of patients' perspectives and experiences. Theories in the domains of social and economic welfare have argued the potential value of patients' perspectives in health research,[21] claiming how a patient's direct testimony of the experienced illness and received healthcare might be too positive when compared with the public's supposedly more objective opinion. The concept of adaptive preferences explains how healthcare, which is formally perceived as substandard, might be perceived as sufficient or good by the receiving individual at that time. We argue, though, partly in line with Nussbaum's philosophy on this, that any patient's testimony, at any time and any place, is valid in and of itself.[21 22] Furthermore, following a human rights-based approach to maternal healthcare, we believe it to be quintessential, in the pursuit of universal high-quality care for all, to include women and their communities in maternal health research and policy making.[23]

### Strengths and limitations

This study involved a wide range and large number of patients from the maternal population in Zanzibar's main hospital, which deals with a high volume of non-complicated and mildly complicated cases as well as with, in its role of referral hospital, the most complicated cases. Nevertheless, it is not a complete representation of the patient population, leading to selection bias. We could not include women who had not reached the hospital, including those who may suffer most from financial barriers. The same applies to the exclusion of patients who were not reachable by mobile phone, which possibly excluded those suffering more from poverty and possibly having less influence in their households and in decision-making processes regarding health. The (majority of) interviews took place in the hospital at the risk of information bias, due to hesitation to criticise the facility they were still in and, sometimes, depending on. Furthermore, we cannot exclude moderator and reporter bias, which we tried to mitigate through a well-defined methodology and interview structure.

Within the group of participants who commented critically beyond the set interview questions, we reached data saturation, thus trusting their comments to reflect a daily experienced reality and consequently providing policymakers with starting points for healthcare improvement.

Methodological challenges led to a percentage of the MNM during the study period to not have been interviewed and for a lower number of control participants to have been included. The final large number of included participants, however, makes us believe that this has not led to selection bias.

### CONCLUSION

This study assessed the perception of patients on their journey to and during maternal care in Zanzibar's main

health facility. First-hand testimonies show that most women seek care in time, access MMH quickly and are treated in time. Nevertheless, a significant minority of them experiences barriers, mostly financial, in reaching care, women with an MNM more so than controls. Quality of care is valued as high and is similar for women with and without an MNM. Simultaneously, a group of participants shared more critical perceptions. There seems to be an opportunity to distinguish between normative and personal perceptions and, thus, combining quantitative and qualitative methodologies should be maintained. Further improvement of maternal health and healthcare will be possible when policies are based on such research, while being patient-centred and setting-specific.

**Acknowledgements** We thank Zuwena Ali Rashid, Vreni Bron, Suhaila Salum Yussuf, Sigrid Hilgen, Marloes Plender, Lisa Trommelen, Jenna Uijt de Haag, Jasmijn Rake, Jasmijn Broerze, Ibrahim Shaban, Elleke van der Meij, Eline Veenstra and Claire Bausch for their enthusiastic, tireless contributions to the data collection. We thank all health workers of Mnazi Mmoja Hospital's departments of Obstetrics & Gynaecology, Internal Medicine, Surgery and Intensive Care for facilitating and supporting participant inclusion and data collection. We thank the administration of Mnazi Mmoja Hospital for facilitating and supporting this study. We are deeply grateful to all the participants for opening up and sharing with us their opinions and ideas.

**Contributors** TH, TM and BJ conceptualised the study. TH, KSM, MR and LD conducted the majority of the data collection and were responsible for data integrity. TH, SC and MPHK performed data analysis. TH and TM supervised the research on site. BJ and AF supervised the research off site. TH drafted the manuscript. All authors revised multiple versions of the manuscript. All authors have read and approved the final version of the manuscript.

**Funding** The authors have not declared a specific grant for this research from any funding agency in the public, commercial or not-for-profit sectors.

**Competing interests** None declared.

**Patient consent for publication** Not required.

**Ethics approval** This research was approved by Zanzibar's Medical Research and Ethics Committee (ZAMREC/0002/JUN/17) and conducted in line with the principles of the Declaration of Helsinki.

**Data availability statement** Data are available on reasonable request. All data relevant to the study are included in the article or uploaded as online supplemental information. All data relevant to the study are included in the article or uploaded as online supplemental information. Deidentified participant data are available from the study's first author (ORCID: https://orcid.org/0000-0003-4513-0063). Reuse is solely permitted after reasonable request with permission from the involved medical ethical committee, the hospital's administration and the study's principal investigator. The study protocol is available on reasonable request.

**ORCID iD**
Tanneke Herklots http://orcid.org/0000-0003-4513-0063

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
