## [Reviewer comments · BMJ Open]

ARTICLE DETAILS

TITLE (PROVISIONAL)	The journey to facility birth in Zanzibar: a questionnaire-based cohort study of patients' perspectives on preparedness, access and quality of care
AUTHORS	Herklots, Tanneke; D'haene, Lara; Mbarouk, Khairat; Rajhy, Mubina; Couperus, Simone; Meguid, Tarek; Franx, Arie; Koster, Maria PH; Jacod, Benoit

VERSION 1 – REVIEW

REVIEWER	Marianne Vidler University of British Columbia, Canada
REVIEW RETURNED	03-Jun-2020

GENERAL COMMENTS	Abstracts - It is not clear what the last sentence of results in the abstract is trying to say. What do you mean by 'critical comments'? I suggest providing more clarification or to remove.- Line 50 – experiences should be passed tense – 'A minority experienced'- Line 51 – update to past tense – 'Quality of facility care was generally highly rated'- The conclusion is not well stated. The recommendation should reflect the study objective, whereas, currently is seems to be a general methodological recommendation. Review lines 67-69, it is not clear There seems to be a strange comma before the end of every sentence with citations, this should be removed. Introduction - Line 77 – remove 'of'- Lines 77-79 – 'limited preparedness' by who? Patients? Families? Health care workers? Facilities? Please specify- Move the study objective from (lines 84-86) to the end of the paragraph- Lines 95-97 – inconsistent reference style Methods - Lines 37-39 – I don't understand what is trying to be said here, what are 'comments of critical nature'? please revise. This statement also seems to reflect a result rather than a method- Table 3 – why are some results bolded? They do not seem to reflect those that are significant Discussion
---

	 - There are many instances where the term 'suboptimal' is used, this is a bit of a strange word, please review and limit its use. - Lines 279-288; this section is vague and not easy to follow, please review. - Be careful with some of the interpretation of findings, such as lines 291-295. I don't believe, you cannot say whether costs were perceived to be higher in cases of maternal near miss because they were in fact higher or maternal near miss where more likely to occur as a result of this barrier to care seeking. This is an important distinction as it has different implications re: your recommendation of preparedness. - The discussion focuses heavily on the use of mixed methods; however, the objective of the study was to report on patient perspectives re: quality of care. Please review the discussion and add more reflection and recommendations related to the study intention.
--	--

REVIEWER	Dr Florence Mgawadere Liverpool School of Tropical medicine , United Kigdom
REVIEW RETURNED	11-Jun-2020

GENERAL COMMENTS	An interesting study to improve quality of care both at facility and community level, however the study does not bring in an new knowledge. Authors need to address the following:  - There is no clear research question for the study - No clear justification for choosing perspectives of MNM as opposed to all pregnant women- there are many studies in literature including Tanzania about patients perspectives on preparedness, access and quality of care - There are no substantive details on how the the controls were selected in the methods section, could authors describe the criteria used to describe controls as "women with uncomplicated and mildly complicated pregnancy". - It is not very clear on how the qualitative data was embedded in the quantitative study, there a slight mention in the methods section and quotes in the results section - Conclusion, it is already known that studies looking at impressions of patient's perspectives and experiences require us of both quantitative and qualitative methods to complement each other
--

VERSION 1 – AUTHOR RESPONSE

Reviewer: 1

- It is not clear what the last sentence of results in the abstract is trying to say. What do you mean by 'critical comments'? I suggest providing more clarification or to remove. > changed to additional comments; the explanation is that in these comments the participants are critical towards the availability of and provision of care
- Line 50 – experiences should be passed tense – 'A minority experienced' > adjusted
- Line 51 – update to past tense – 'Quality of facility care was generally highly rated > adjusted
- The conclusion is not well stated. The recommendation should reflect the study objective, whereas, currently is seems to be a general methodological recommendation. > agreed and adjusted accordingly, see line 153 till the end of the paragraph; adjusted in parallel in the conclusion of the corpus
- Review lines 67-69, it is not clear > adjusted

- There seems to be a strange comma before the end of every sentence with citations, this should be removed. > this is the journal's required way to place the references in the text
- Line 77 – remove 'of' > adjusted
- Lines 77-79 – 'limited preparedness' by who? Patients? Families? Health care workers? Facilities? Please specify > adjusted
- Move the study objective from (lines 84-86) to the end of the paragraph > adjusted
- Lines 95-97 – inconsistent reference style > in accordance with the journal's standard of referring to an unpublished piece of work
- Lines 37-39 – I don't understand what is trying to be said here, what are 'comments of critical nature'? please revise. This statement also seems to reflect a result rather than a method > we rephrased; you are right that this is a result, but it did lead us to perform additional analysis, which we otherwise would not have done, hence the mention of this in the methods section. Do you think it is adequate to leave it there or should we transfer it to the results section?
- Table 3 – why are some results bolded? They do not seem to reflect those that are significant > in tables 1, 2 and 4, the bolded results are the significant outcomes; in table 3, this does not apply, so we removed the bolding
- There are many instances where the term 'suboptimal' is used, this is a bit of a strange word, please review and limit its use. > good point, we reviewed and rephrased accordingly
- Lines 279-288; this section is vague and not easy to follow, please review. > revised accordingly
- Be careful with some of the interpretation of findings, such as lines 291-295. I don't believe, you cannot say whether costs were perceived to be higher in cases of maternal near miss because they were in fact higher or maternal near miss where more likely to occur as a result of this barrier to care seeking. This is an important distinction as it has different implications re: your recommendation of preparedness. > agreed, we have revised, line 324-328
- The discussion focuses heavily on the use of mixed methods; however, the objective of the study was to report on patient perspectives re: quality of care. Please review the discussion and add more reflection and recommendations related to the study intention. > we revised and feel that the study intention and discussion are aligned now

Reviewer: 2

An interesting study to improve quality of care both at facility and community level, however the study does not bring in an new knowledge. Authors need to address the following:

- There is no clear research question for the study > we attempted to rephrase/clarify this accordingly in line 118-120
- No clear justification for choosing perspectives of MNM as opposed to all pregnant women- there are many studies in literature including Tanzania about patients' perspectives on preparedness, access and quality of care > added paragraph in introduction (lines 90-96, 120-122)
- There are no substantive details on how the controls were selected in the methods section, could authors describe the criteria used to describe controls as "women with uncomplicated and mildly complicated pregnancy". > rephrased, line 130-131
- It is not very clear on how the qualitative data was embedded in the quantitative study, there a slight mention in the methods section and quotes in the results section > we added sentences in the method section to explain this better, line 148-151
- Conclusion, it is already known that studies looking at impressions of patient's perspectives and experiences require us of both quantitative and qualitative methods to complement each other > this is true indeed, nevertheless, the specific setting of Zanzibar, in which limited maternal health research has been done, allowed for a seminal study to be performed this way. It has really highlighted how the statistics, often extrapolated from or based on national or other regional statistics, cannot and will not show the full extend of the burden of disease. Albeit we are not claiming to have new insights, we do show how these also apply to this study's population. It hopefully will motivate and inform local research agendas as such.

We hope to have sufficiently addressed your feedback. We look forward to hearing back from you.

VERSION 2 – REVIEW

REVIEWER	Marianne Vidler University of British Columbia, Canada
REVIEW RETURNED	23-Sep-2020

GENERAL COMMENTS	Thank you for the opportunity to review this manuscript again. This is an important study and contributes meaningfully to the literature. The manuscript still requires a thorough edit by a native English speaker, there are several awkwardly worded sentences that are hard to follow and should be reworked to improve the flow and impact of the paper, some of these are noted below. line 76 – ‘are not timely prepared’ is awkward – reword line 88 – define MNM abbreviation line 92, 94, 96, 98, 103 – weird punctuation at end of sentence - . . Line 100-102 – weird reference style Line 107 – remove ‘allow to effectively’ How were the controls matched? Were they the next delivery that matched your criteria or chosen at random? What do you mean by ‘similar’ date of admission, mode and gestational age? Can you be more exact? +/- how many days/hours/weeks? Line 127 – replace ‘circumstances’ with characteristics Line 133-135 – “Considering the latter two metrics, ...” take a look at this sentence, it is not clear as currently written. Line 175 – replace ‘received’ with completed or attended Lines 297-299 – simplify this sentence, it is unnecessarily wordy and hard to follow Lines 310-311 – I would replace ‘MNM’ with ‘women who experienced a near miss complication’, these women are more than their complications so it is important to first describe them as women and not ‘outcomes’. The interpretation section of the discussion should include comments related to how these findings fit with what is known in the literature and other research regarding access to care in cases of near miss. This section should also address why you did or did not find differences between women with a near miss complication and those without. The results are mentioned but they are not explained, other than cost. Lines 326-327 – rework this sentence Line 348 – replace ‘pursue’ with ‘pursuit’
--

REVIEWER	Florence Mgawadere Liverpool school of Tropical Medicine,
REVIEW RETURNED	21-Sep-2020

GENERAL COMMENTS	Relevant piece of work on MNM with good command of English and insight on the limitations Could the authors explain in the method section why they used a quantitative method to answer a research question looking into insights of women? other researcher already recommend use of qualitative research to answer research questions. Be explicit on the description of the controls- it is not clear at present
--

	The findings include Quotes from women- it will be good to indicate in the method section how these quotes were extracted- The results section is presented like the authors used mixed method. Interpretation of the results inadequate lacks detailed synthesis of the main findings I
--	---

VERSION 2 – AUTHOR RESPONSE

To Reviewer 1

- Could the authors explain in the method section why they used a quantitative method to answer a research question looking into insights of women? other researcher already recommend use of qualitative research to answer research questions.

> Employing quantitative measurements of experiences around child birth and satisfaction with health care are not restricted to our study. Recently, efforts to design validated standard outcome measurements sets, such as the ICHOM Pregnancy and Childbirth set, have also incorporated quantitative measurements using a Likert scale just as in our case. The main advantage of a quantitative approach is the possibility to compare across settings, countries and over time. We do, however, agree with the reviewer that close attention should be given to the coherence between quantitative and qualitative results. Our results show that, in the case of Zanzibar's main hospital, the discrepancy between quantitative measurements of quality of care and patient experiences and findings from qualitative interviews is too large to justify implementation of a quantitative-only method in the future by policy makers.

- Be explicit on the description of the controls- it is not clear at present > adjusted
- The findings include Quotes from women- it will be good to indicate in the method section how these quotes were extracted- > we added this information
- The results section is presented like the authors used mixed method. Interpretation of the results inadequate lacks detailed synthesis of the main findings
> adjusted accordingly

To Reviewer 2:

Comments to the Author

Thank you for the opportunity to review this manuscript again. This is an important study and contributes meaningfully to the literature.

The manuscript still requires a thorough edit by a native English speaker, there are several awkwardly worded sentences that are hard to follow and should be reworked to improve the flow and impact of the paper, some of these are noted below. > thank you for this advice, we acted accordingly and hope you will also feel the improvements

line 76 – ‘are not timely prepared’ is awkward – reword > done

line 88 – define MNM abbreviation > it was used in line 85 already, so adjusted that accordingly

line 92, 94, 96, 98, 103 – weird punctuation at end of sentence - . . > this is the reference style of the journal

Line 100-102 – weird reference style > this is the reference style of the journal for non-published work

Line 107 – remove ‘allow to effectively’ > done

How were the controls matched? Were they the next delivery that matched your criteria or chosen at random? What do you mean by ‘similar’ date of admission, mode and gestational age? Can you be

more exact? +/- how many days/hours/weeks? > adjusted accordingly

Line 127 – replace ‘circumstances’ with characteristics > done

Line 133-135 – “Considering the latter two metrics, ...” take a look at this sentence, it is not clear as currently written. > rewritten

Line 175 – replace ‘received’ with completed or attended > done

Lines 297-299 – simplify this sentence, it is unnecessarily wordy and hard to follow > done

Lines 310-311 – I would replace ‘MNM’ with ‘women who experienced a near miss complication’, these women are more than their complications so it is important to first describe them as women and not ‘outcomes’. > although I very much have a similar sentiment to this as the reviewer, we use the phrase maternal near-miss OR woman/patient who had/survived a near-miss complication so often, that I we do feel it would take up a lot of text to write it out each time. We have now attempted in our introduction and discussion to describe it fully, and in the result section to be more concise. We have made a note in the introduction that we will refer to women who have suffered from the complications as MNM in the continuum of the manuscript.

The interpretation section of the discussion should include comments related to how these findings fit with what is known in the literature and other research regarding access to care in cases of near miss. This section should also address why you did or did not find differences between women with a near miss complication and those without. The results are mentioned but they are not explained, other than cost. > adjusted accordingly

Lines 326-327 – rework this sentence > adjusted

Line 348 – replace ‘pursue’ with ‘pursuit’ > done

We hope you find that we have been able to process your feedback adequately and to your satisfaction. I do feel the manuscript improved significantly from your input so I genuinely want to thank you for that.